# Comparative Efficacy of Common Active Ingredients in Organic Insecticides Against Difficult to Control Insect Pests

**DOI:** 10.3390/insects11090614

**Published:** 2020-09-08

**Authors:** Galen P. Dively, Terrence Patton, Lindsay Barranco, Kelly Kulhanek

**Affiliations:** Department of Entomology, University of Maryland, College Park, MD 20742, USA; tpatton@umd.edu (T.P.); lbarranc@umd.edu (L.B.); kkulhane@umd.edu (K.K.)

**Keywords:** organic insecticide, control efficacy, spinosad, pyrethrin, azadirachtin

## Abstract

**Simple Summary:**

According to USA organic standards, farmers can apply a certified allowable insecticide when all non-chemical practices fail to control pests. However, there exists a lack of control efficacy information to enable decision-making about which organic product works best for a given target pest. In this study, we conducted 153 field trials on different host crops to evaluate the control efficacy of common active ingredients in organic insecticides against insect pest groups considered difficult to control in organic production. The performance of organic products Entrust (spinosad), Azera (pyrethrin and azadirachtin), PyGanic (pyrethrin) and Neemix (azadirachtin) varied widely among pest groups, as well as among pest species within a group, providing an overall reduction in pest infestations by 73.9%, 61.7%, 48.6% and 46.1%, respectively. Those insect pests that were particularly difficult to control included thrips, stinkbugs, cucumber beetles and fruitworms. Several caveats pertaining to the application of the results are discussed.

**Abstract:**

There exists a lack of control efficacy information to enable decision-making about which organic insecticide product works best for a given insect pest. Here, we summarize results of 153 field trials on the control efficacy of common active ingredients in organic insecticides against 12 groups of the most difficult to control insect pests. These trials evaluated primarily the organic products Entrust (spinosad), Azera (pyrethrin and azadirachtin), PyGanic (pyrethrin) and Neemix (azadirachtin), which reduced pest infestations by an overall 73.9%, 61.7%, 48.6% and 46.1% respectively, averaged across all trials. Entrust was the most effective control option for many insect pests, particularly providing >75% control of flea beetles, Colorado potato beetle, cabbageworms and alfalfa weevil, but was relatively ineffective against true bugs and aphids. Azera provided >75% control of green peach aphid, flea beetles, Japanese beetle, Mexican bean beetle, potato leafhopper and cabbageworms. PyGanic was less effective than Entrust and Azera but still provided >75% control of green peach aphid, flea beetles and potato leafhopper. The growth inhibition effects of azadirachtin in Neemix were particularly effective against larvae of Mexican bean beetle and Colorado potato beetle but was generally less effective in trials with insect infestations consisting mainly of adult stages. Those insect pests that were particularly difficult to control included thrips, stinkbugs, cucumber beetles and fruitworms. Several caveats pertaining to the application of the results are discussed.

## 1. Introduction

Organic production in the U.S. has experienced phenomenal growth since the 1990s, with double-digit increases in the number of certified farms during most years and current production accounting for 5.8% of total food sales in 2019 [1]. In dealing with pest management, organic farmers are challenged with the same insect pests confronting conventional farmers; however, they must rely first on a system-based use of biological, cultural, mechanical and physical practices to reduce or avoid pest problems. When these practices fail to control pests, the National Organic Program Standards [2], NOP Code 205.206 (e), allows farmers to apply a biological, botanical or synthetic substance on their organic crops. The Organic Materials Review Institute [3] provides a comprehensive list of 380 generic and brand name insecticide products allowed for organic use according to the NOP rule.

Three major categories of insecticides widely used in organic production include products formulated with spinosad, pyrethrin and neem derivatives. Collectively, these active ingredients are contained in about one third of the insecticidal products allowed for organic use. Spinosad is a natural product composed of a mixture of spinosyns A and D fermentation metabolites of a soil-dwelling actinomycete, *Saccharopolyspora spinosa* [4]. It has broad-spectrum activity against many major lepidopteran pests, thrips, leaf miners and certain beetle species, and is an active ingredient in several conventional insecticides. Spinosad disrupts binding of acetylcholine in nicotinic acetylcholine receptors at the postsynaptic cell of the insect, leading to involuntary muscle contractions, tremors and paralysis [5]. Pyrethrins are extracted from dried flowers and seeds of the pyrethrum plant (*Chrysanthemum cinerariifolium*). Organic formulations have the same mode of action as the conventional pyrethroid insecticides, functioning as a sodium channel modulator by disrupting the impulses along the axons of neurons, resulting in paralysis and death of the insect [6]. Pyrethrin has broad spectrum of activity on many insects, causing a quick knockdown as a contact and stomach poison, but breaks down rapidly in sunlight. Derivatives from the leaves and seed of the neem tree, *Azadirachta indica* A. Juss (Meliaceae), have been used for centuries for medical and pesticidal purposes and are currently available in various organic formulations of oils, soaps and extracts containing mainly the compound azadirachtin [7]. Neem-based products have a very broad range of behavioral and physiological effects on many groups of insect pests, acting as an antifeedant, insect growth regulator (IGR), repellent, sterilant and inhibitor of oviposition [8,9,10]. Depending on the formulation and application method, neem constituents can also be absorbed through plant roots and leaves [11,12,13,14], providing some level of systemic activity on certain insect pests.

Organic farmers clearly have many insecticide products at their disposal; however, managing insect pests with these products can be a difficult challenge for several reasons. Organic insecticides are relatively short-lived, many degrade rapidly under environmental conditions, and thus require frequent applications, precise timing and sufficient knowledge to use them properly. Most products are more effective on the immature stages of insects; however, because they are used mainly as a management solution of last resort, timing applications at the most vulnerable insect stages to achieve maximum control efficacy is often not possible. Additionally, organic insecticides are mainly stomach and contact poisons with limited systemic toxicity, so thorough spray coverage on plants is essential to ensure direct exposure to the target pest for effective control. Lastly, organic insecticides are much more expensive compared to conventional products, so farmers need to know whether investing in insecticide control will actually result in an economic gain. Regarding this issue, there is a general lack of control efficacy information for organic farmers to decide which insecticide product works best for a given target pest. A number of organic pest management guides provide control efficacy rankings of available organic insecticides against insect pests [15,16,17,18,19]. However, much of the available information is categorical and based on a limited number of published field trials, technical reports and personal communications.

Here, we summarize the results of 153 field trials conducted during 2002 to 2015 to evaluate the control efficacy of commonly used organic insecticides. Trials focused mainly on the organic products Entrust™ (spinosad, Corteva Agriscience, Indianapolis, IN, USA), Neemix^®^ 4.5 (azadirachtin, Certis USA, Columbia, MD, USA), PyGanic^®^ (pyrethrins, McLaughlin Gormley King Company, Minneapolis, MN, USA) and Azera^®^s (pyrethrins and azadirachtin, McLaughlin Gormley King Company, Minneapolis, MN, USA), representing the three active ingredient categories described above. The comparative efficacy of these insecticides against different insect pests is not well studied. Other organic insecticides and experimental products were also tested in most trials. We combined the percent control data across trials to show the overall performance and range of control efficacy of each insecticide against 12 of the most difficult to control pest groups.

## 2. Materials and Methods

### 2.1. General Trial Design

Field evaluations started in 2002 and continued with multiple trials each year until 2015. Most testing was conducted at the Central Maryland Research and Education Facility, Beltsville, MD, USA, but some trials were located at the Wye Research and Education Facility, Queenstown, MD, USA, and Central Maryland Research and Education Facility, Upper Marlboro, MD, USA, where certain pest populations were present to evaluate treatments. We established trials on non-certified organic land and applied standard conventional fertility and weed management inputs depending on the specific requirements of the host crop. Plots were planted either as transplants or direct seeded at normal planting times (spring and summer crops: May to early June, fall crops: August) either on black plastic mulch or bare ground seedbeds. We applied irrigation, as needed, using drip tape or overhead systems. Treatments were arranged in all trials as a randomized block design, replicated four times. Each plot ranged from a single row 6 m long to three rows 7.5 m long. Row width varied from 0.75 to 1.8 m according to the planting system and specific crop. No seed or foliar insecticides were applied, except for the treatments being tested. However, certain crops were planted with fungicide-treated seed.

### 2.2. Insect Pest/Crop Groups

Multiple trials were conducted each year during 2002 to 2015 on different host crops (Table 1). We focused on 12 pest groups that are considered difficult to control in organic production, as indicated in several farmer surveys [20,21] and identified by Caldwell et al. [15] as important pests that lack organic insecticide efficacy data. For certain pest groups, different host crops were used in trials for a given year and across years.

### 2.3. Treatments

We mainly focused on evaluating treatments of Azera, Neemix, PyGanic and Entrust; however, not all trials tested these insecticides together. Azera (McLaughlin Gormley King Company, Minneapolis, MN; active ingredients (a.i.) azadirachtin at 1.2% and pyrethrins at 1.4% by volume) was tested as an experimental formulation during 2002 to 2007; after which, Azera was officially registered in 2008. Azera was mainly applied at the commonly used rate of 2.34 L/h, but several trials included rates from 1.17 to 4.68 L/h. Neemix 4.5 (Certis USA, Columbia, MD, USA; AI azadirachtin at 4.5% by volume) was applied at the 1169 mL/h rate in most trials, except a few also tested the lower rate of 585 mL/h. PyGanic Crop Protection EC 1.4 (McLaughlin Gormley King Company, Minneapolis, MN, USA; AI pyrethrins at 1.4% by volume) was mainly applied at the rate of 2.34 L/h, although several trials included a range of rates from 1.17 to 4.68 L/h. Entrust (Corteva Agriscience, Indianapolis, IN, USA; AI spinosad) was initially tested in earlier trial years as an 80 W formulation (80% a.i. by weight), followed by the liquid formulation (2SC Naturalyte at 22.5% a.i. by weight) during later years. These formulations were applied at the standard 71.2 g a.i./h rate for most pest groups, but several trials included rates ranging from 52.7 to 105.5 g a.i./h.

We also evaluated mixtures of the insecticides with various additives and other organic and experimental products. Those products evaluated for control of specific pest groups included: Gemstar^®^ (Certis USA, Columbia, MD, USA; 0.64% polyhedral occlusion bodies of the nuclear polyhedrosis virus of *Helicoverpa zea*), *Bacillus thuringiensis* (Bt) foliar products (Deliver^®^, Agree^®^ WG, and Javelin^®^ WG, Certis USA, Columbia, MD, USA), Aza-Direct^®^ (Gowan, Yuma, AZ, USA; 1.2% azadirachtin), Venerate™ (Marrone Bio Innovations, Davis, CA, USA; 94.46% heat-killed *Burkholderia* spp. strain A396), PFR 97^®^ (Certis USA, Columbia, MD, USA; 20% *Isaria fumosorosea* Apopka Strain 97), Trilogy^®^ (Certis USA, Columbia, MD, USA; 70% extracts of neem oil), M-Pede^®^ (Gowan, Yuma, AZ, USA; 49% potassium salts of fatty acids), Des-X^®^ (Certis USA, Columbia, MD, USA; 47% potassium salts of fatty acids), Ecotec^®^ (Brandt Consolidated, Inc., Springfield, IL, USA; contains several essential oils), Ultra-Pure^®^ Oil (BASF, Research Triangle Park, NC, USA; 98% mineral oil), Surround^®^ WP (Engelhard Corporation, Iselin, NJ, USA; 95% kaolin clay), and adjuvants Nu-Film^®^ P (Miller Chemical, Hanover, PA, USA), BioLink^®^ Spreader-Sticker (Westbridge, Vista, CA, USA) and Oroboost^®^ (Oro Agri, Fresno, CA, USA).

We applied all insecticides as foliar treatments using a CO_2_ backpack sprayer with different boom arrangements and nozzle types depending on the plot width and canopy structure of the crop. We used a 3 m boom with six flat fan nozzles in trials with multiple row plots, or a 0.9 m boom with three hollow cone nozzles covering a single row plot, with one nozzle dropped on each side of the crop canopy and one over the top. The depth of the drop nozzles on each side varied depending on the height of the crop canopy. The backpack sprayer was calibrated to deliver 188 to 357 L/h of diluted insecticide spray depending on the boom size and crop canopy. In most trials, we first applied treatments when the density of the target pest population was determined high enough to cause economic damage. Different criteria for making this determination depended on the pest/crop group. Each treatment was repeated on a weekly schedule for a variable number of applications to allow for a relative evaluation of control efficacy among different insecticides (more details are given in the Results and Discussion Section).

### 2.4. Insect Sampling

In all trials, we made general observations prior to the first application to assess the relative infestation level of the target pest. Following each application, we sampled plots at least twice, usually at 2 and 6 days post-treatment, in order to assess the level of pest density. Sampling focused on the major insect pest; however, we recorded other minor pests and beneficial arthropods in many trials. Different sampling methods and sample sizes were used depending on the pest/crop group to measure changes in insect densities after each treatment. For cabbageworms, Mexican bean beetle, cucumber beetles, flea beetles, Japanese beetles, Colorado potato beetle, Harlequin bugs, squash bugs and green peach aphid, we visually examined 6 to 12 consecutive plants from the center of each plot and recorded the number of insects per plant. Pea aphid and alfalfa weevil densities were estimated by taking 10 sweeps per plot using a standard 38 cm sweep net. Depending on the host crop, either sweep net or vacuum sampling over a portion of each plot assessed the density of adult potato leafhoppers, while samples of 10 excised leaves per plot were examined to estimate numbers of leafhopper nymphs. For stinkbugs on tomato and pepper, we harvested mature fruit weekly from each plot and recorded the number of damaged fruits. Onion thrips density was determined by counting the number of adults and immatures per onion plant, whereas the number of flower thrips on zinnia plants were extracted and recorded from samples of 10 to 20 flower heads per plot using Berlese funnels. In many trials, we also collected data on the marketable yield per plot, percentage of defoliation over the entire plot and percentage of damaged plants.

### 2.5. Data Analysis

We analyzed data from each trial as a randomized block analysis of variance (ANOVA) using SAS Proc Mixed [22] to test for insecticide effects on the target pest population. The model included insecticide treatment, sampling date and the interaction as fixed factors, replicate block as a random effect and sampling date as a repeated measure. Before analysis, we tested the raw data for normality and homogenous variance using the Shapiro–Wilk *W* test, Spearman’s rank correlation and by examining residual scatter plots. We performed data transformations prior to analysis, and partitioned the variance if necessary. The interaction effect was not significant in most trials. However, when it was significant, differences among treatments changed in magnitude but remained relatively ranked in the same order across sampling dates. In both cases, we averaged the post-treatment data on insect density over all sampling dates of each trial and calculated the mean percent control for each insecticide, using Abbott’s formula [(control density-treatment density)/control density ×100]. We then pooled the percent control means of all trials for each pest/crop group to represent the overall control efficacy of the insecticide. This pooled dataset only included data from trials with consistent pest densities high enough to rigorously test for treatment effects. The pooled results for each insecticide were visualized with box-whisker plots to display the percent control data of individual trials, the 25% and 75% percentile range of the trial data and the overall mean control efficacy. We calculated the 95% confidence limits (95% CL) around the mean using a Student’s t distribution to indicate significant differences in overall control efficacy among insecticides tested for each pest/crop group.

## 3. Results and Discussion

### 3.1. Thrips

Thrips are major pests causing direct and indirect damage to many field and greenhouse-grown crops worldwide [23]. In organic production, they are particularly challenging to manage due to their high generation turnover, low aesthetic injury threshold tolerated on many crops and their ability to transmit several plant pathogens [24,25]. We evaluated organic insecticides in eight and ten trials during different years on onion and zinnia (grown for cut flowers), respectively. Because both crops were greenhouse-grown and then transplanted in early spring, plants became quickly infested during early May, primarily due to immigrating thrips from nearby senescing small grain fields. On onions, infestations consisted primarily of onion thrips, reaching mean densities of adult and larval stages ranging from 11.5 to 56.5 per plant in untreated plots. Infestations in most trials caused significant injury characterized by silvery patches or streaks on the leaves. On zinnia, Eastern flower thrips along with several other flower species infested the flower heads. Populations in untreated plots ranged from 2.3 to 62.5 thrips per head and caused discoloration of the flower petals and the failure of some buds to open. Depending on the trial, two or three weekly treatments of each insecticide were applied per trial, starting at the first sign of thrips activity.

Results of thrips control are summarized in Figure 1 by crop because insecticide treatments performed differently due to the species complex and variable levels of insecticide exposure. Overall, control efficacy of all treatments was slightly better but more variable against flower thrips (42.1%) compared to control of onion thrips (32.0%). However, relative differences among insecticides were similar for both crops. Entrust at the standard rate reduced thrips’ densities by an overall 65.6% in both crops, which was significantly higher than the other treatments based on non-overlapping 95% CL. In some trials, treatments with Entrust plus Oroboost (2% v/v) or M-pede (1% v/v) consistently improved control by another 10% to 15%. Azera, Neemix and PyGanic averaged 29.3%, 19.1% and 20.7% over all trials, respectively. Doubling the standard rates of Azera and PyGanic or adding adjuvants Nu-Film P or BioLink showed some enhanced efficacy but levels of control were still considerably lower than Entrust.

We measured onion yield in four trials but only one showed a significant yield increase in the Entrust plots. Moreover, there was no significant relationship in any trial between thrips numbers and yield; thus, the level of control was apparently not high enough for treatments to show a yield response or the onion plants compensated for the injury. Variation in control among trials, particularly for Azera, Neemix and PyGanic, was noticeably greater for flower thrips. In several trials, densities in treated plots averaged higher than levels in untreated plots. The variable results were most likely due to differences in species composition, ratio of adults to larvae and disproportionate levels of insecticide coverage on flower heads. Several studies reported that the direct contact and systemic activity of neem-based formulations resulted in high mortality of early larvae but had no effect on older larvae and adult thrips [26,27]. In conventional production of cut flowers, multiple insecticide applications are often needed to manage thrips because they feed in tight, protective places within the expanding flower structure [28]. No signs of phytotoxicity were evident with any of the treatments.

Our results agree with other studies reporting 42% to 62% control of onion and flower thrips with Entrust, and generally ineffective control (<30%) with products containing pyrethrin, azadirachtin or combinations of both [29,30,31,32,33,34,35,36,37,38,39].

### 3.2. Aphids

Aphids are common pests of most major families of organic crops [40]. Their piercing-sucking feeding removes plant sap, causing yellowing and curling leaves, stunted growth and deformed fruits. As they feed, aphids exude sticky honeydew on leaves, which encourages the growth of sooty mold. Certain species also transmit plant virus diseases that reduce the marketable quality of the crop or can actually kill plants. We conducted 12 trials to evaluate the control efficacy of Azera, Neemix and PyGanic against aphid infestations on several host crops.

Figure 2 summarizes the results separately for trials infested predominantly with green peach aphid (GPA) and pea aphid (PA). For GPA, we evaluated each insecticide in five trials of mixed plantings of cabbage and collards. Cabbage aphids (*Brevicoryne brassicae*) were also present in several trials but densities were not high enough to discern treatment effects. GPA infestations averaged 23.3 per plant and ranged up to 58 per plant in untreated plots. Each insecticide was applied weekly either two or three times depending on the trial. Overall, control efficacy averaged 78.3%, 52.6% and 75.1% for Azera, Neemix and PyGanic respectively, but overlapping 95% CL indicated no significant differences. Three trials in 2006 showed a small but consistent increase in control efficacy with treatments of PyGanic applied with different adjuvants (Nu-Film P, BioLink). A weekly treatment schedule of Entrust at the standard rate was also tested against GPA in two collard trials and gave <40% control.

For PA, we applied a single application of each insecticide in seven alfalfa trials. Untreated infestations ranged from 3.5 to 11.5 aphids per sweep, which was below the reported economic threshold of 30 aphids per sweep [41]; however, densities were consistently high enough to test for treatment effects. Overall, percent control averaged 19.4%, 16.0% and 20.3% for Azera, Neemix and PyGanic, respectively. Together, these insecticides provided significantly less control of PA compared to GPA control, and the range of control among trials was greater, except for Neemix. None of the organic treatments were significantly different from the untreated control, and higher rates of Azera showed no consistent pattern of control enhancement compared to the standard rate. Moreover, treatments of Azera mixed with the adjuvant Oroboost (4% v/v) showed only a small improvement in PA control compared to Azera applied alone. In contrast to the GPA trials, the lower PA control was most likely due to having only a single application per trial. Azadirachtin works primarily as a feeding inhibitor and IGR against aphids [42,43], thus, evaluations of Azera and Neemix control at 2–3 days after a single application may be too early to reveal the cumulative effects on immature aphids. Furthermore, PA populations have developed resistance to a number of conventional insecticides, including pyrethroids, which may partially explain the low control efficacy of PyGanic and Azera.

We evaluated other organic products and treatment combinations for aphid control in several trials. A greenhouse trial in 2008 tested several soap and oil products against a heavy infestation of GPA on pansy bedding plants. Compared to the control, Ultra-Pure oil at 2% v/v, M-Pede at 2% v/v, Trilogy at 2% v/v and Oroboost at 4% v/v provided 62.5%, 42%, 34% and 28% control, respectively. No signs of phytotoxicity were evident with any of these treatments. Oroboost at 4% v/v and Ecotec at 1.17 L/h, when each were applied alone in two alfalfa trials, provided no control of PA. In a 2015 soybean trial, two weekly applications of Des-X at 2% v/v reduced a high infestation of soybean aphid (*Aphis glycines*) by 47%, but control was significantly less than the PyGanic treatment.

Findings of other researchers show a similar range of control efficacy for azadirachtin and pyrethrin products against GPA, flower aphids and other related species. Reported levels of aphid control range from 64% to 92% with Azera [44,45,46], 19% to 47% control with Neemix [47,48,49,50] and 47% to 73% with PyGanic [44,51,52]. Although Entrust provided ineffective GPA control in our trials, several studies show moderate levels of control efficacy ranging from 41% to 64% against several aphid species [53,54]. Field trials testing the control efficacy of pyrethrin combined with azadirachtin against PA generally showed some suppression (<30%) but not enough to provide effective control [55,56]. Our results on PA are consistent with these findings.

### 3.3. True Bugs

True bugs are at the top of the list of difficult pests to control in organic production. Their piercing-sucking feeding causes wilting and stunting of plants, premature abortion of fruiting bodies, damaged fruit and unmarketable leafy produce of many horticultural crops [57,58,59,60,61]. Eleven trials were conducted on the following true bugs: harlequin bug (HB) on collards (6 trials), squash bug (SB) on zucchini (4 trials) and stinkbugs (STB) on tomato and bell pepper (7 trials). Two or three applications were applied in most trials, except for those evaluating STB, which required 5 to 6 weekly applications to cover the fruiting period of tomato and pepper. Infestations per plant in untreated plots ranged from 0.5 to 7.1 HB and 2.1 to 37.2 SB. We did not directly sample stinkbug populations but instead measured treatment effects by the percentage of fruit showing feeding injury, which ranged from 9.4% to 92.7% of the marketable crop in untreated plots.

Results in Figure 3 are summarized separately for each true bug, which are known to exhibit differences in insecticide susceptibility among species [62]. Differences among insecticide treatments were relatively the same for HB and SB, although the overall level of control was 20% lower for the latter bug. Overall, Azera, Neemix and PyGanic reduced HB densities by 62.1%, 38.3% and 71.0%, and SB densities by 49.2%, 33.7% and 52.2%, respectively. However, percent control among treatments was not statistically different based on overlapping 95% CL, except for PyGanic, which gave significantly higher control of HB compared to the other insecticides. These results are somewhat consistent with other field studies that showed a wide range of control efficacy with organic insecticides against true bugs. One study [63] reported no effective control of HB with Neemix and PyGanic but Entrust at the higher rate reduced nymphs by 72% and adults by 58% on collards and turnips. Leaf dip bioassays also showed toxicity levels of spinosad higher than levels for pyrethrins and azadirachtin [64,65]. Another trial reported only 17% and 30% control of HB with Entrust and Neemix, respectively [66]. For SB, results of the few published trials show some control of nymphal densities with pyrethrin and azadirachtin formulations but insufficient control of adults under high population pressure [60]. However, Seaman et al. [67] reported 74% control of SB with Azera but at the highest labeled rate.

Overall, reductions in STB-damaged fruit resulting from Azera, PyGanic and Entrust treatments averaged 35.6%, 9.3% and 32.8%, respectively (Figure 3). Higher rates of Azera and Entrust generally provided additional control but still less than 60% protection from fruit injury. Even with five weekly applications of each insecticide, control was consistently ineffective, because the majority of fruit injury was caused by more tolerant adults. Compared to other true bugs, these results indicate that adult STB are less susceptible to Azera and PyGanic, since we applied nearly twice the number of applications.

To further evaluate control efficacy against brown marmorated stinkbug nymphs, we manually infested plots in a 2013 pepper trial with 18 nymphs (75% 2nd–3rd, 25% 4th–5th instars) per plant, starting when crown fruit reached marketable size. After five days to allow nymphs to acclimate, we applied two treatments three days apart of each insecticide alone and in combination with other products. Treatments that significantly reduced nymphal densities included Azera at 2.34 L/h (79.2%), PyGanic at 4.68 L/h (73.3%) and Entrust at 71.2 g a.i./h (84.5%). Treatments that were less effective included M-Pede 2% v/v (46.8%) and PFR 97 at 1.68 kg/h (22.7%). We also did not observe any significant gain in nymphal control with Entrust or Azera in combination with M-Pede.

Other studies have reported variable control efficacy of STB with organic insecticides, which are consistent with our findings on Azera, PyGanic and Entrust. Laboratory bioassays have shown that pyrethrin, pyrethrin combined with azadirachtin, spinosad, azadirachtin, and insecticidal soaps have activity against STB [68,69,70]. However, field studies are not always consistent with laboratory results. Several studies found no significant reduction in STB densities with these same active ingredients compared to the control [53,59,71], generally reporting <35% control. Only three field studies reported effective control of STB with organic insecticides: 70% control with Azera on soybean [59], 84% control with Entrust on snap bean [45] and 82% control with Neemix on cowpea [72]. Taken together, the wide range of true bug control is likely due to differences in the ratio of adult and nymphs, population density, number of applications and residue coverage influenced by the spray volume and canopy structure of the specific crop. Overall, our findings, along with other published reports, confirm that adult true bugs are difficult to control with organic insecticides; however, there is still the opportunity to effectively control nymphs if applications are timed properly.

### 3.4. Flea Beetles

Flea beetles are common pests of many cruciferous and solanaceous crops. They are difficult to control with organic insecticides because they feed on more protected parts of the plants and can quickly recolonize fields after treatment [73]. Feeding injury results in numerous small holes in the leaves of seedlings and transplants, causing stunting or even plant death, and reduced marketable yield of leafy produce [74]. We evaluated flea beetle control during the early seedling stages in 11 eggplant trials. Several species of flea beetles (mainly eggplant and tobacco in order of abundance) invaded plots shortly after transplanting in each trial, after which insecticide treatments were applied either 2 or 3 times on a weekly basis. Overall infestations in untreated plots averaged 6.2 flea beetles per plant with levels ranging up to 21 beetles per plant.

The standard rates of Azera, PyGanic and Entrust significantly reduced flea beetle populations by an overall 73.6%, 76.6% and 77.4%, respectively (Figure 4). Although results suggest a rate response for Azera, individual pairings of different rates within trials showed only slight increases in control with higher rates of Azera. Likewise, treatments of PyGanic mixed with adjuvants (Nu-Film P, BioLink) provided no significant increase in control compared to PyGanic applied alone. Neemix was significantly less effective, averaging 31.9% reduction in flea beetle numbers. Azadirachtin apparently had some repellent or antifeedant activity against colonizing flea beetles but expectedly, did not have any IGR effect on adults. For this reason, pyrethrin in PyGanic and Azera was probably the main cause of flea beetle mortality. We also evaluated Surround WP as a physical barrier and direct irritant applied in combination with 1% v/v Trilogy in several trials and significantly reduced flea beetle numbers by 75%. However, the addition of Surround with Azera provided little additional control compared to Azera applied alone. We found no significant differences in plant height and yield among treatments, but plants in control plots were generally smaller and produced fewer fruit. This effect may be related to the moisture stress caused by feeding injury during early seedling growth.

Our findings align closely with other field studies testing organic materials against different species of flea beetles. Taken together, most studies show consistently high levels of flea beetle suppression with Entrust and PyGanic, but limited efficacy with azadirachtin products [64,75,76,77,78,79].

### 3.5. Cucumber Beetles

Cucumber beetles colonize fields shortly after seedlings emerge or are transplanted and feed on the cotyledons and young leaves, which either kills plants or greatly retards their growth [80]. More importantly, overwintered beetles carry the causal agent of bacteria wilt, *Erwinia tracheiphila*, which is transmitted to susceptible cucurbits as beetles feed on young plants. These beetles are particularly difficult to manage with short-residual organic insecticides because they quickly re-invade after treatment and are often sheltered from insecticide sprays during the day in soil cracks or under the straw or plastic mulch.

We evaluated organic insecticides in four cucumber trials and two zucchini trials that were direct-seeded in fields that had a history of cucumber beetle activity the previous year. Trials received two or three weekly applications, starting a few days after plant emergence when beetles were active. In all trials, moderate to high populations of striped cucumber beetles caused significant leaf and cotyledon injury, with several trials showing bacterial wilt symptoms during later plant growth. Infestation densities in untreated plots ranged from 1.6 to 3.1 beetles per seedling. Pooled over trials, the overall reduction of beetles relative to the control averaged 48.7% for Azera, 46.1% for Neemix, 39.4% for PyGanic and 56.2% for Entrust (Figure 5). All treatments prevented plant stand losses due to direct feeding injury but did not provide enough control of adults in most trials to prevent spread of bacterial wilt. Because beetles quickly re-colonized treated plants, multiple insecticide treatments applied at shorter intervals may be necessary for effective control and prevention of bacterial wilt.

Laboratory bioassays have documented that permethrin can cause high mortality of striped cucumber beetles and extracts of neem have antifeedant effects on adults [81,82]. However, our findings and those of other field trials consistently show only low to moderate levels of control (<50%) with organic insecticides [19,67,76,83,84,85]. Given the effectiveness of current insecticides available to organic farmers, it is very difficult to prevent the feeding injury and spread of bacterial wilt by these insects.

### 3.6. Japanese Beetles

Japanese beetles (JB) cause defoliation injury to over 300 plant species and can significantly reduce the marketable quality or aesthetic value of many horticultural crops [86]. We evaluated the effectiveness of single applications of Azera, Neemix and PyGanic on two susceptible specialty crops (5 trials on basil, 3 trials on marigold flowers). Since other studies have shown inconsistent control of JB from direct contact toxicity, we focused on the residual activity of the insecticides as a repellent and feeding deterrent. In all trials, JB were active throughout the treatment period, reaching densities ranging from 2 to 8 beetles per plant and causing >20% bloom or foliage injury on untreated plants.

Trial results of percent control at two and four days post-treatment in Figure 6 indicate the residual activity of the treatments. Azera, Neemix and PyGanic reduced JB densities at two days post-treatment by an average 85.2%, 77.5% and 62.1%, respectively. At four days post-treatment, control efficacy of Azera and Neemix dropped by an overall 29% but still significantly reduced the number of JB per plant by 57.1% compared to untreated controls in most trials. The residual activity of PyGanic dropped even more at four days, averaging 30.1% control. Azera treatments were consistently more effective and showed longer residual activity at the higher rates, providing >90% control at two days and 71% to 88% at four days. Because control performances of Azera and Neemix were not significantly different, and PyGanic was considerably less effective, the repellency and antifeedant effects of azadirachtin accounted for most of the control efficacy. Doubling the rates of PyGanic or adding adjuvants (Nu-Film P, BioLink) consistently increased the level of control compared to a standard rate of PyGanic applied alone. Two trials on marigold plants also tested Entrust and showed an average 50.2% control of JB when applied alone or 77.5% when mixed with 1% M-Pede after five days post-treatment. None of the treatments caused phytotoxic symptoms.

Other studies have reported evidence of residual repellency and feeding deterrent activity on JB with foliar applications of azadirachtin [87,88,89,90]. Vitullo and Sadof [91] tested foliar and soil applications of azadirachtin for control of JB on roses. They reported a 55% reduction in rose bloom injury with weekly applications of azadirachtin but the level of bloom protection was still unacceptable. Soil applications of neem extracts have also shown variable results for control of JB grubs [81,92,93,94] but generally not enough suppression of adult emergence to prevent plant injury. Although the cut flower trials show some evidence of Entrust efficacy, other studies on raspberries have reported poor control with spinosad [95,96]. Our results indicate that azadirachtin in Azera and Neemix can suppress JB activity but only for a relatively short period; thus, it may require multiple treatments under high population levels to protect susceptible organic crops from these highly mobile adults.

### 3.7. Mexican Bean Beetle

The Mexican bean beetle (MBB) ranked ninth in importance out of 29 problem insects identified by organic farmers [97]. Overwintered adults begin feeding on the earliest planting of snap beans soon after seedlings emerge, and then populations increase through multiple generations on successive plantings, resulting in the highest infestations in late plantings [98,99]. Both larvae and adults feed on the underside of leaves, causing reduced photosynthetic activity and desiccation of the plant. Infestations during pod development can reduce yield if the amount of defoliation exceeds 20% [100,101].

We conducted nine trials in different years on late plantings of snap beans. To attract a high infestation, we established an earlier untreated plot at each trial location to serve as a nursery for MBB population buildup. In all trials, adult beetles invaded plots during the early vegetative stages and larval infestations reached damaging levels prior to bloom. Percent defoliation in untreated plots ranged from 28% to 73%, and late larval densities ranged from 5 to 17 per plant. Each treatment was applied weekly either 4 or 5 times depending on the trial, starting as soon as the first egg masses hatched.

Figure 7 shows separate trial results for reductions in defoliation and late larvae (3rd and 4th instars) relative to the untreated control plots. Averaged over trials, treatments of Azera and Neemix significantly suppressed densities of late larvae by 79.6% and 81.7% respectively, and both insecticides reduced the level of defoliation by 90%. However, reductions in larval densities did not occur until after two applications of each insecticide, but eventually, the overall cumulative effects resulted in 80% to 90% fewer larvae reaching pupation in most trials. The IGR activity of azadirachtin in both insecticides had a major effect on larval development and reduced feeding due to secondary physiological effects. Several trials showed no consistent gain in control efficacy with higher rates of Azera. In contrast, Azera applied at the lower rate of 1.17 L/h and Neemix at the lower rate of 585 mL/h provided the same level of control as the standard rates. These results suggest that the cumulative effect of the IGR activity was not dependent on the application rate. Bean yields in five trials showed significant increases of 20% to 38% in the Azera and Neemix treatments compared to yields in the untreated control.

The knockdown and direct toxicity effects of PyGanic resulted in significantly less control against larvae (52.2%) but still reduced defoliation by an overall average of 73.5%. In several trials, treatments of PyGanic with different adjuvants (Nu-Film P, BioLink) showed a slight gain in control but not significantly different from PyGanic alone. Entrust was tested in only four trials and showed similar but more variable levels of control efficacy (53.4%). We also pooled data on adults recorded in seven trials, which showed mean control efficacy ranging from 12% to 83% for Azera, 10% to 82% for Neemix and 10% to 68% for PyGanic. Effects on adults were more variable due to inter-plot movement during the treatment period of each trial. Other tested organic products included Surround WP applied in combination with 1% v/v Trilogy, which reduced late larvae densities by 24.8% and defoliation by 39.3%. Similarly, combining Surround with Azera provided very little additional control compared to Azera alone. Aza-Direct, another azadirachtin product, applied at 2.34 L/h, reduced levels of late larvae and defoliation that were similar to those provided by Neemix.

There is considerable research published on controlling MBB with conventional insecticides and nonchemical approaches [100]; however, few studies have reported on the control efficacy of organic insecticides tested in these trials. Seaman [16] lists 27 organic insecticides labeled for MBB control but indicated that only eight products containing azadirachtin alone or in combination with permethrin were considered effective. One study by Nottingham and Kuhar [101] reported 57% control with the high rate of Azera, 39% control with PyGanic and 49% control with Entrust, when each insecticide was applied twice weekly.

### 3.8. Potato Leafhopper

Potato leafhopper (PLH) is major pest of potatoes and legumes, such as snap beans, dry beans and alfalfa, but also infests other horticultural and field crops [102]. Nymphs and adults feed by piercing and sucking plant tissues, causing a blockage of the plant vascular tissue, which increases plant respiration and reduces photosynthesis. Injury symptoms include yellowing and curling of leaf margins, leading to eventual leaf necrosis (hopperburn). PLH is difficult to manage with nonchemical approaches in organic production due to its wide host range, high dispersal behavior and few effective natural enemies.

We evaluated Azera, Neemix and PyGanic for PLH control in four trials of snap beans, five trials of alfalfa and six trials of potato. Treatments were applied weekly for an average of 1.6, 2.7 and 2.5 applications on alfalfa, potato and green beans, respectively. Since Entrust is not labeled for PLH and previous trials have indicated poor control, we tested the standard rate in only three trials. Treatments commenced when adults began to invade plots or when adult densities exceeded one per sweep, depending on the crop. PLH infestations reached moderate to high levels in most trials, causing noticeable yellowing and leaf curl injury by the end of the treatment period. Average densities in untreated plots ranged from 1.1 to 6.2 adults per sweep or 1.3 to 6.4 nymphs per leaf, depending on the trial.

Trial results for adults and nymphs are given separately in Figure 8. Overall, control by Azera, Neemix and PyGanic averaged 80.5%, 25.3% and 68.1% for adults, and 85.0%, 44.0% and 79.5% for nymphs, respectively. Trials testing different rates of Azera showed a consistent rate response, with higher rates up to 4.68 L.h^−1^ averaging 87.6% control compared to 76.9% control at the standard 2.34 L/h rate. The addition of the adjuvant Oroboost with Azera resulted in no improvement in control. In several trials, PyGanic with different adjuvants (Nu-Film P, BioLink) also showed only slight gains in control and not significantly different from PyGanic alone. Control efficacy of Azera and PyGanic was mainly due to the knockdown and direct mortality effects of pyrethrin on adults and nymphs, while the repellency and antifeedant effects of Neemix provided poor control of adults and only moderate but highly variable efficacy against nymphs. Neemix performed better in trials with earlier infestations of developing nymphs that allowed more time for the IGR effect to work during the treatment period.

Of the three trials testing Entrust, control was consistently less than 10%; however, control efficacy increased to 75.2% when Entrust was applied in combination with 1% v/v M-Pede in one trial. PyGanic formulations alone or in combination with Neemix (equivalent to Azera) have provided satisfactory control of adults and nymphs in other trials, while Neemix alone has been ineffective [15,103,104].

Other types of organic products evaluated for PLH control were only moderately effective. In several trials, M-Pede at 2% v/v, Oroboost at 4% v/v, Ecotec at 1.17 L/h and Des-X at 2% v/v applied alone reduced PLH numbers by 18.1%, 32.4%, 30.8% and 38.6%, respectively. These findings are in general agreement with published results on leafhopper control but provide a quantitative assessment of the range in control efficacy of these insecticides.

### 3.9. Colorado Potato Beetle

Colorado potato beetle (CPB) can cause complete loss of a potato crop if not controlled and inflicts economic damage to eggplant when infestations are high. Control is very challenging, even for conventional farmers, owing to the insect’s ability to develop resistance to insecticides. Seaman et al. [17] recommend nine management options for organic control of CPB, including some dire measures, such as flaming, vacuum collecting and trench trapping. If these management options fail, spinosad and azadirachtin products have proven to be effective rescue treatments in most situations.

We conducted 14 potato trials to evaluate different organic insecticides and treatment combinations against CPB. Treatments were initiated when the first-generation population exceeded 40 small larvae per 10 plant hills, and either 2 or 3 weekly applications of each insecticide were applied depending on the trial. Populations of CPB were consistently high in all trials, causing 36.6% to 73.3% defoliation and up to 53% tuber yield loss in untreated plots. Although all stages were sampled, densities of adults, egg masses and early instars (1st and 2nd instars) were highly variable and did not show any significant treatment effect in most trials. Older larvae (3rd and 4th instars) caused the majority of feeding injury, and thus reductions in their density and resulting defoliation provided better indicators of the control effectiveness of each insecticide. Numbers of older larvae in untreated plots ranged up to 22.2 per plant hill.

Figure 9 shows the individual trial results of each insecticide expressed as the percent reduction in larval densities relative to the untreated control. The direct toxicity effects of Entrust at the 71.2 g a.i./h rate provided the most effective and consistent control, significantly reducing larval densities and defoliation by greater than 95%. Additionally, we tested a range of Entrust application rates in several trials and found that lower rates from 14.3 to 29.3 g a.i./h also provided good control, ranging from 70% to 88% reduction of older larvae. These findings suggest that one or two weekly applications of Entrust at reduced label rates may provide effective control of CPB.

In comparison, Azera (71.4% control) and Neemix (63.3% control) were less effective but still gave relatively good control of late larvae at the standard rate. The feeding deterrent and IGR effects of azadirachtin had a greater impact on larval development than was evident by the observed reductions in larval numbers, because both insecticides reduced defoliation by about 86%. Such reduction in leaf damage would be acceptable economic control in organic potato production because plants can normally withstand up to 30% defoliation without yield loss. Neem extract products inhibited feeding of adult and larval CPB in laboratory studies [105,106,107] but are generally slower to bring about control in the field. Compared to our results, other field studies reported similar control effectiveness with Entrust (>95% control) and Azera (68% to 77% control) [103,108,109,110,111]. Nault and Seaman [108] also found no significant difference in control between low and high application rates of Entrust.

PyGanic was the least effective treatment and more variable across trials, reducing larval densities and defoliation by only 14.7% and 34.3%, respectively. CPB is known to exhibit resistance to pyrethroid insecticides [112], which is probably attributed to the poor control by permethrin. In several trials, applications of PyGanic at higher rates and in combination with adjuvants and synergists gave only small insignificant increases in control.

### 3.10. Cabbageworms

Imported cabbageworm (ICW), cabbage looper (*Trichoplusia ni*) and diamondback moth (DBM) are common insect pests of cole crops. Mixed infestations of these caterpillars can cause extensive feeding injury on the leaves and head of cabbage, broccoli, cauliflower and other crucifers, resulting in contaminated and unmarketable produce [113,114]. Organic insecticides are widely used as a rescue treatment to protect cole crops against these pests. Seaman [18] lists 27 organic products for cabbageworm control but acknowledges that only a few evaluated in trials show greater than 50% effectiveness.

We evaluated different organic insecticides in spring and fall plantings of cabbage (9 trials), collards (5 trials) and broccoli (5 trials). ICW was the predominant cabbageworm, comprising 70% of the infestations with densities ranging from 0.6 to 23.3 larvae per plant in untreated plots. Lower DBM densities ranged from 0.7 to 3.6 larvae per plant but were still consistently high enough to discern treatment differences. Feeding injury caused significant reductions in head development and subsequent >50% losses in marketable yield in untreated plots. Because ICW was the predominant pest, we applied the first application when adult butterflies were active and eggs were hatching. Each insecticide was then repeated weekly either 3 or 4 times depending on the trial. Control efficacy was indicated by the percent reduction in cabbageworm density and the percent of marketable plants at the end of the treatment period. Broccoli heads were marketable if no caterpillars and feeding injury were present, while cabbage heads and collard plants were marketable if only the outer leaves had minor damage. Since treatment differences were relatively the same for the three crops, we combined the efficacy data on each insecticide across all trials.

Figure 10 presents trial results for each caterpillar which responded differently to the treatments. Entrust applied at the 71.2 g a.i./h rate provided consistent residual activity against both cabbageworms in all trials, averaging 92.2% control of DBM and 96.1% control of ICW. Treated plots yielded 98% to 100% marketable cabbage/broccoli heads and collard plants at harvest compared to less than 20% in untreated plots. Entrust provided significantly higher control efficacy than the other treatments based on non-overlapping 95% CL. Azera was also consistently effective in reducing densities of DBM and ICW by an average of 79.4% and 85.5%, respectively. The higher 4.68 L/h rate of Azera resulted in only a small increase in control compared to the 2.34 L/h rate. Although Azera reduced larval densities less in most trials compared to Entrust, 96% to 100% of the cabbage heads in Azera-treated plots were still marketable according to organic standards. The combined effects of permethrin and azadirachtin apparently prevented larvae from developing to older instars that bore deeper into the cabbage head below the wrapper leaves.

Control effectiveness of Neemix and PyGanic was more variable and significantly lower compared to Azera and Entrust. Moreover, both insecticides were less effective against DBM (overall 48.6% control) compared to ICW (overall 67.2% control). Percent of marketable plants in Neemix and PyGanic-treated plots ranged from 59% to 83% compared to less than 20% in untreated plots. DBM has a long history of developing resistance to many insecticide classes, including pyrethroids [113]; thus, resistance may have contributed to the low effectiveness of permethrin in PyGanic. Furthermore, azadirachtin has minimal contact activity, so Neemix is most effective as an effective IGR when ingested by larvae. DBM larvae feed primarily on the underside of leaves, and adult moths avoid treated surfaces [115,116]; thus, larvae were probably less exposed to insecticide-treated surfaces. No signs of phytotoxicity were evident with any of the treatments.

We also evaluated other organic products in a number of trials. Agree WG (841g/h) and Javelin WG (1.7 kg and 3.4 kg/h) provided similar levels of control in four trials, averaging 78% and 82% reduction of DBM and ICW respectively, and resulting in 71% to 84% marketable cabbage and broccoli heads. Bt insecticides have been widely used to control lepidopteran larvae but are relatively short-lived and more effective on early instars. Nevertheless, the control efficacy of the Bt treatments was about the same as Azera. Treatments of Surround WP plus 1% Trilogy reduced cabbageworm densities by an average of 51.3% across three trials. Our results corroborate the findings on cabbageworm control by other field studies that reported 85% to 99% control with spinosad-based insecticides [117,118,119,120], 84% to 91% control with Azera [121,122], 47% to 76% control with Neemix and other azadirachtin products [122,123] and 67% to 80% control with Bt products [120,124,125].

### 3.11. Alfalfa Weevil

Several curculionid species are major pests causing feeding injury to fruiting structures, leaves and roots of organic crops. However, information on the control efficacy of organic insecticides against these pests is very limited. Given the increased demand for organic dairy feed [126], we evaluated the effectiveness of different rates of Azera, Neemix, PyGanic and Entrust against the alfalfa weevil (AW), which represents a reasonable surrogate for other weevil species. Ten trials were conducted on separate years in alfalfa fields that were at least two years old. Treatments were initiated during late April when leaf tip injury was first observed. Infestations consisted mainly of young larvae when treatments were applied, averaging 26 per sweep in untreated plots. Trials received one application of each insecticide, except for a few trials with two applications a week apart.

Entrust at the 71.2 g a.i./h rate reduced larval populations by an overall 86.8% compared to the other insecticides that provided <30% control (Figure 11). Doubling the rate of Entrust resulted in 96% control but this higher rate may not be cost-effective for a lower-value forage crop. Although low rates of Entrust were not tested, alfalfa can tolerate some damage, so lower rates could possibly reduce the AW population to a tolerable level. Other studies also provide evidence that spinosad is the most effective organic control option for AW and other curculionid species. Reddy et al. [127] evaluated six commercially available biorational insecticides against AW larvae under laboratory conditions and reported 100% mortality with Entrust within three days post-treatment. In field studies, Entrust, at 71.2 g a.i./h, provided 95% control of AW [54] and 77% control of pepper weevil [128].

The higher rates of Azera and PyGanic showed some additional suppression but still not enough to be effective control options. Neemix was also relatively ineffective, yet laboratory studies have shown high mortality, repellency and IGR effects of azadirachtin extracts against AW and other curculionids [127,128,129,130,131]. However, our findings agree with field studies on pepper weevil that show >30% control efficacy with Neemix [132,133,134,135,136]. In our trials, the antifeedant and IGR properties of azadirachtin in Azera and Neemix would likely provide better control if multiple treatments were applied over a longer period. However, such a treatment regime may be more expensive than a single application of Entrust.

### 3.12. Fruitworms

Several types of lepidopteran larvae are major pests of fruit and vegetable crops in organic production. The predominant species is the corn earworm (CEW) (or tomato fruitworm) that feeds internally near the stem end of the tomato and pepper, creating a messy, watery cavity contaminated with excrement and cast skins [137]. The damaged tissue becomes infected with molds, and injured fruits often rot before harvest. On sweet corn, young CEW hatching from eggs laid on silks quickly work their way down the silk channel into the ear where they feed on the kernels at the ear tip [137,138]. Even with conventional insecticides, management of fruitworms on these host crops is problematic due to their dispersal capability to recolonize fields after treatment, narrow window of opportunity to apply insecticides and difficulty in obtaining adequate spray coverage and residual activity.

We evaluated the control efficacy of different organic insecticides against fruitworms in nine trials on trellised tomato plantings and six trials on sweet corn. All trials were planted later in the growing season to increase the chance of higher infestations. On tomato, mixed infestations consisted of CEW, yellow-striped armyworm and variegated cutworm, together causing an average 18.6% of the fruit damage in untreated plots across all trials. Insecticide treatments were initiated at the beginning of fruit set and repeated weekly five times to cover the fruiting period in each trial. At 6 days following each treatment, we harvested all ripe fruit to record the number of damaged fruits.

Figure 12 displays the individual results of the tomato trials expressed as overall percent reduction in damaged fruit for Azera (59.4%), Neemix (51.9%), PyGanic (33.4%) and Entrust (72.3%). Entrust at the standard rate provided the most effective control but performance varied across trials and was not significantly different from the other insecticides according to non-overlapping 95% CL. PyGanic was the least effective, showing a wide range of control and no consistent gain in effectiveness when mixed with different adjuvants (Nu-Film P, BioLink). Azera and Neemix treatments gave moderate levels of fruit protection due mainly to the antifeedant and IGR effects of azadirachtin, but results were also variable across trials. Several trials tested low and high application rates of Azera but found no significant differences from the standard rate. Several factors contributed to the variation across trials. Particularly, control efficacy was dependent on how well the weekly applications coincided with susceptible larval stages of the different fruitworm species. Canopy density of the trellis plants also varied across trials, which probably influenced the level of spray penetration and residue coverage. Since most trials experienced moderate infestation levels and received five weekly treatments over the entire fruiting period, it may be difficult to achieve acceptable control of tomato fruitworms with these insecticides under high population pressure.

In two trials, four weekly applications of Deliver at 1.1 kg/h resulted in 92.8% reduction in fruit damage, and five weekly applications of Gemstar at 731 ml/h significantly reduced CEW-damaged fruit by 89.5%. However, fruitworm infestations were well below economic levels in both trials. In general, our results agree with other field studies that report reductions in fruitworm damage ranging from 35% to 78% for Entrust, 65% to 78% for Bt products and 42% to 68% for Gemstar [128,139,140,141,142]. Neem products applied alone and in combination with pyrethrins have been evaluated to a lesser extent for fruitworm control. Several earlier studies reported unsatisfactory control with neem seed oil [143,144], while one study reported 46% control with Neemix [145].

In sweet corn trials, treatments were applied at 25% to 50% silking and repeated on a 2-, 3- or 4-day schedule depending on the trial and moth activity. The number of applications per trial ranged from five to nine treatments. We sampled 50 ears per plot at fresh market maturity to determine the percentage of ears damaged. In all trials, heavy CEW infestations caused significant kernel injury to 50% to 95% of the ears in control plots. Due to differences in spray schedules, treatments tested and CEW population pressure, we did not combine control efficacy data across trials.

Overall, Entrust, applied at the standard 71.2 g a.i./h rate, provided the most effective control of CEW compared to the other treatments. Since Entrust was approved for organic use in 2003, other studies have reported effective control with Entrust [146,147,148,149], and fact sheets on organic sweet corn recommend Entrust as the most effective option for CEW control [15,150]. However, our results show that ear protection with Entrust varied depending on the spray interval and number of applications. Of the different spray schedules tested, percent reduction in ear damage averaged: 15% in 2002 (5 sprays 3 to 4 days apart), 33% in 2004 (4 sprays 3 days apart), 73% in 2005 (7 sprays 2 to 3 days apart), 92% in 2006 (9 sprays 2 days apart), 32% in 2008 (7 sprays 3 days apart) and 68% in 2009 (6 sprays 3 to 4 days apart). These results demonstrate that timing applications is critical, especially during the silking period that may take 3 to 4 days for silks to emerge from all ears. Spinosad is broken down rapidly by sunlight [151] and thus may have only a few days of residual activity on treated silk tissue. Besides, treatments applied during early silk will not protect ears that expose silks later. Thus, treatments of Entrust applied at relatively short intervals during silking (such as in the 2006 trial) are required under high CEW pressure to achieve adequate ear protection for organic quality sweet corn.

Of other products tested, Azera, Neemix and PyGanic were relatively ineffective compared to Entrust. The Bt product, Deliver, applied alone and mixed with vegetable oil injections in the silk tube, also gave unsatisfactory control in a 2002 trial [152], which agrees with other studies that have shown inconsistent results with the ear tip injection technique, when CEW pressure is high [153,154,155]. Because Bt insecticides require ingestion and degrade rapidly in sunlight, trials testing foliar sprays have also reported unsatisfactory control of lepidopteran larvae in sweet corn [147,148,156].

## 4. Conclusions

Our results compiled over multiple trials during different years provide the first quantitative assessment of the range of control efficacy by Entrust, Azera, PyGanic and Neemix against the most difficult to control insect pests in organic production. The performance of these insecticides varied widely among pest groups, as well as among pest species within a group. Weighted for the number of trials and averaged across all pest groups, Entrust, Azera, PyGanic and Neemix reduced pest infestations by an overall 73.9%, 61.7%, 48.6% and 46.1%, respectively. Entrust applied at the 71.2 g a.i./h rate of spinosad provided >75 % control of flea beetles, Colorado potato beetle, cabbageworms and alfalfa weevil. Nearly as effective against CPB larvae were lower rates of Entrust compared to the standard rate. Entrust also provided better control than the other insecticides against thrips, cucumber beetles, fruitworms and corn earworm in sweet corn, reducing infestations by >50%. As expected, Entrust was relatively ineffective against sucking insects, particularly true bugs and aphids. Although trial results showed that Entrust was the most effective control option for many insect pests, field-evolved resistance to spinosad has occurred in a number of insect pests, especially diamondback moth, several thrips species and Colorado potato beetle [157]. For example, there is evidence of high-level CPB resistance to spinosad on organic potato farms, after repeated overuse of Entrust [158,159]. For this reason, whenever possible, organic farmers should consider using other classes of insecticides in rotation with Entrust to reduce the risk of resistance development.

The combination of azadirachtin and pyrethrin in Azera at the rate of 2.34 L/h ranked second in overall performance, providing >75% control of green peach aphid, flea beetles, Japanese beetle, Mexican bean beetle, potato leafhopper and cabbageworms. Azera tested at higher rates or at shorter spray intervals provided 10% to 20% additional control of true bugs, flea beetles, Japanese beetle and potato leafhoppers. Azera was generally more effective against pest infestations comprised of both adult and immature stages, which allowed for the combined direct toxicity, behavioral and physiological effects of both active ingredients. Pyrethrin in PyGanic at the 2.34 L/h rate was less effective but still provided overall >75% control of green peach aphid, flea beetles and potato leafhopper. In general, higher rates of PyGanic or the addition of adjuvants did not consistently increase control of most pest groups. The repellency, antifeedant and growth inhibition effects of azadirachtin in Neemix provided >75% control against Mexican bean beetle and >60% control against Colorado potato beetle larvae, even at the lower rate. Arguably, Neemix would be more effective against pest groups in infestations consisting mainly of developing immature stages. However, adults were the predominate stage in most trials and thus subject only to the repellent and antifeedant effects of azadirachtin.

There are some noteworthy caveats when applying these findings. First, organic farmers use insecticides only as a last resort after all nonchemical methods have been explored. This management scenario frequently involves higher infestations of older insect stages compared to the younger age structure of infestations that triggered treatments in our trials. Thus, control efficacy of insecticides applied as a rescue treatment against older infestations may be lower than the levels reported here. Secondly, the spatial scale of our trials was conducive to movement of adults among small plots during the treatment period, which was an issue with highly mobile pests, such as thrips, flea beetles, cucumber beetles and potato leafhoppers. Possibly, the control efficacy of the insecticides could be higher when applied to whole fields, where there is less chance of insects re-invading the treated area from outside or from untreated control plots. Thirdly, we arbitrarily implied above that >75% reduction in the pest population provided acceptable control by the organic insecticide. Caldwell et al. [15] also defines ’good control’ of an organic insecticide as a statistically significant reduction in pest density or damage by 75% or more compared to the untreated control. However, we would argue that lower levels of control may be acceptable to many organic farmers depending on their production goals and quality standards for organic product marketability. For example, several insecticides reduced pest densities by <75%, but more than 90% of the crop was still marketable according to organic standards. Finally, insecticides were applied in all trials when the target pest infestation was high enough to eventually cause economic damage, followed by fixed weekly treatments that were not applied according to any threshold of pest activity. Unquestionably, there were trials on certain pest groups, wherein fewer applications would have been necessary to provide acceptable control. For example, one or two properly timed applications of Entrust or Azera could provide acceptable control of Colorado potato beetle, Mexican bean beetle or cabbageworms. Thus, further investigations should be undertaken to determine the minimum number of applications, spray interval and costs associated for each insecticide regimen in order to provide acceptable control of each pest group. The number of applications would depend on the pest population density, pest susceptibility to the particular insecticide, duration of crop stages vulnerable to pest injury and the benefit/cost ratio of each application. Furthermore, future research should examine how to incorporate organic insecticides with different modes of action into treatment rotations that maximize the direct toxicity and indirect behavioral and physiological effects on insect pests. Although our trials were not designed to evaluate the best management strategy, knowledge of the range and overall control efficacy of each insecticide reported here will help pest management practitioners and organic farmers choose the most effective product to manage each pest group.

## Figures and Tables

**Figure 1 insects-11-00614-f001:**
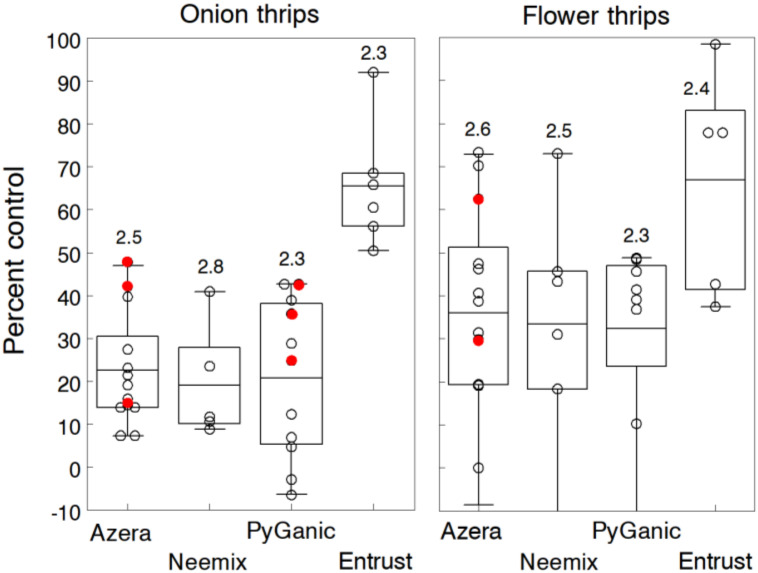
Percent control of onion and flower thrips with weekly applications of Azera, Neemix, PyGanic and Entrust relative to the untreated control. Mean data for each insecticide are given for 8 and 10 trials on onion and zinnia, respectively. Box-whisker plots show the 25% and 75% percentile range of percent control data and the horizontal line in each box is the overall mean control efficacy. Number above each plot indicates the average number of weekly applications. Application rates of Azera, Neemix, PyGanic and Entrust were 2.34 L/h, 1169 mL/h, 2.34 L/h and 71.2 g a.i./h respectively, except for trial data in red indicating a higher rate. Entrust provided significantly higher control of onion thrips compared to the other insecticides. There were no significant differences among treatments for flower thrips.

**Figure 2 insects-11-00614-f002:**
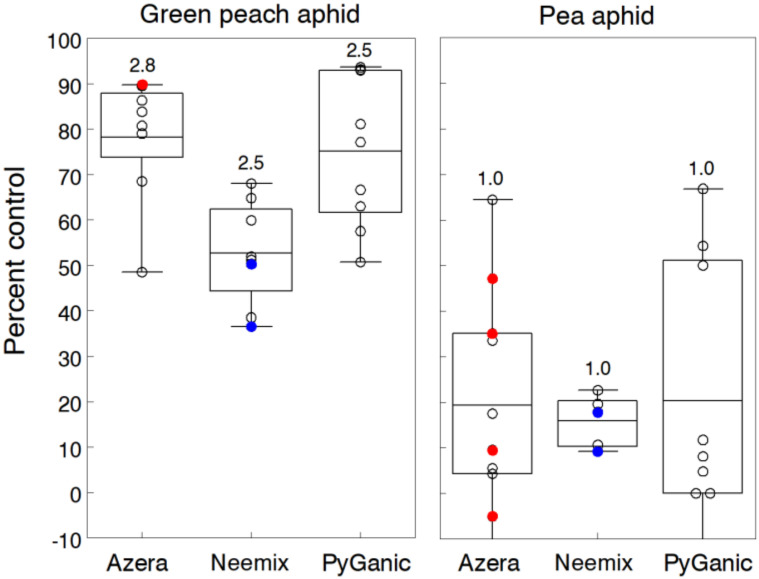
Percent control of green peach aphid and pea aphid with weekly applications of Azera, Neemix and PyGanic relative to the untreated control. Mean data for each insecticide are given for five and seven trials on cole crops and alfalfa, respectively. Box-whisker plots show the 25% and 75% percentile range of percent control data and the horizontal line in each box is the overall mean control efficacy. Number above each plot indicates the average number of weekly applications. Application rates of Azera, Neemix, PyGanic and Entrust were 2.34 L/h, 1169 mL/h, 2.34 L/h and 71.2 g a.i./h respectively, except for individual trial data in blue and red indicating lower and higher rates, respectively. There were no significant differences among treatments for each aphid.

**Figure 3 insects-11-00614-f003:**
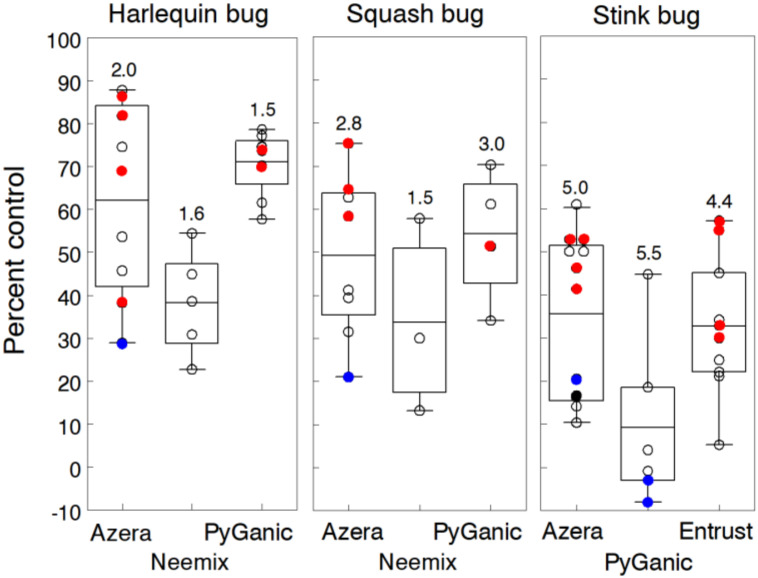
Individual trial results of organic insecticide control of true bugs. Azera, Neemix and PyGanic were evaluated for control of harlequin bug and squash bug, while Azera, PyGanic and Entrust were evaluated against stinkbugs. Mean percent control data for each insecticide are given for six trials with harlequin bug on collards, four trials with squash bug on zucchini and seven trials with stinkbugs on tomato and pepper. Box-whisker plots show the 25% and 75% percentile range of percent control data and the horizontal line in each box is the overall mean control efficacy. Number above each plot indicates the average number of weekly applications. Application rates of Azera, Neemix, PyGanic and Entrust were 2.34 L/h, 1169 mL/h, 2.34 L/h and 71.2 g a.i./h respectively, except for individual trial data in blue and red indicating lower and higher rates, respectively. Insecticide treatments for each true bug were not significantly different, except for PyGanic, which provided significantly better control of harlequin bugs than the other insecticides.

**Figure 4 insects-11-00614-f004:**
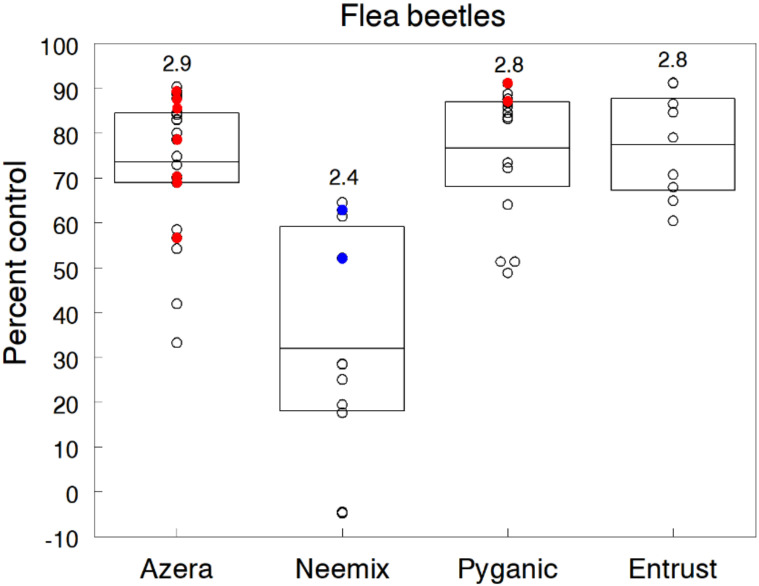
Percent control of flea beetles with weekly applications of Azera, Neemix, PyGanic and Entrust relative to the untreated control. Mean data are given for 11 trials on eggplant. Box-whisker plots show the 25% and 75% percentile range of percent control data and the horizontal line in each box is the overall mean control efficacy. Number above each plot indicates the average number of weekly applications. Application rates of Azera, Neemix, PyGanic and Entrust were 2.34 L/h, 1169 mL/h, 2.34 L/h and 71.2 g a.i./h respectively, except for individual trial data in blue and red indicating lower and higher rates, respectively. Neemix provided significantly lower control than the other insecticides.

**Figure 5 insects-11-00614-f005:**
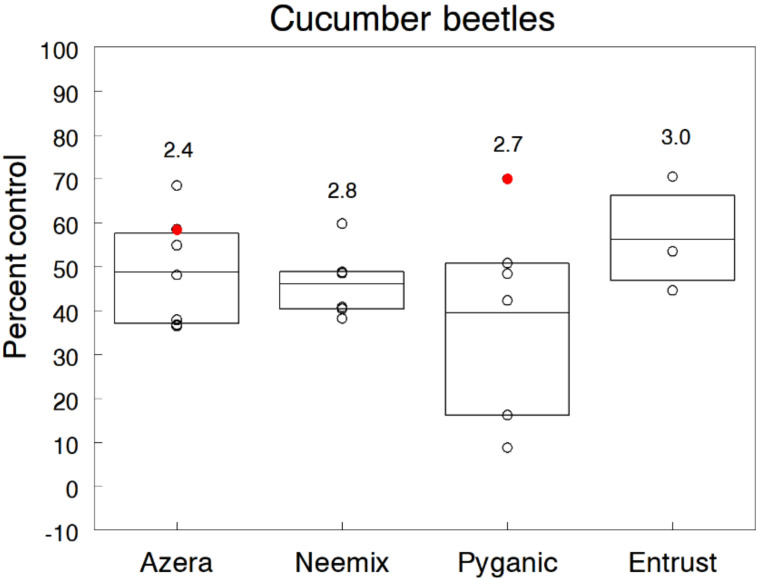
Percent control of cucumber beetles with weekly applications of Azera, Neemix, PyGanic and Entrust relative to the untreated control. Mean data are given in four cucumber trials and two zucchini trials. Box-whisker plots show the 25% and 75% percentile range of percent control data and the horizontal line in each box is the overall mean control efficacy. Number above each plot indicates the average number of weekly applications. Application rates of Azera, Neemix, PyGanic and Entrust were 2.34 L/h, 1169 mL/h, 2.34 L/h and 71.2 g a.i./h respectively, except for trial data in red indicating a higher rate. There were no significant differences among treatments.

**Figure 6 insects-11-00614-f006:**
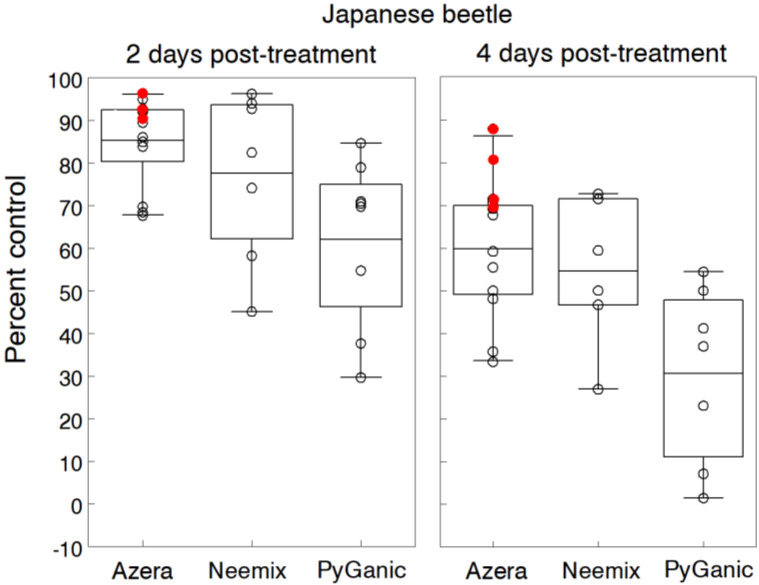
Percent control of Japanese beetles with a single application of Azera, Neemix and PyGanic relative to the untreated control. Mean data at two and four days post-treatment are given for five trials on basil and three trials on marigold flowers. Box-whisker plots show the 25% and 75% percentile range of percent control data and the horizontal line in each box is the overall mean control efficacy. Application rates of Azera, Neemix, PyGanic and Entrust were 2.34 L/h, 1169 mL/h, 2.34 L.h^−1^ and 71.2 g a.i./h respectively, except for trial data in red indicating a higher rate. There were no significant differences among treatments.

**Figure 7 insects-11-00614-f007:**
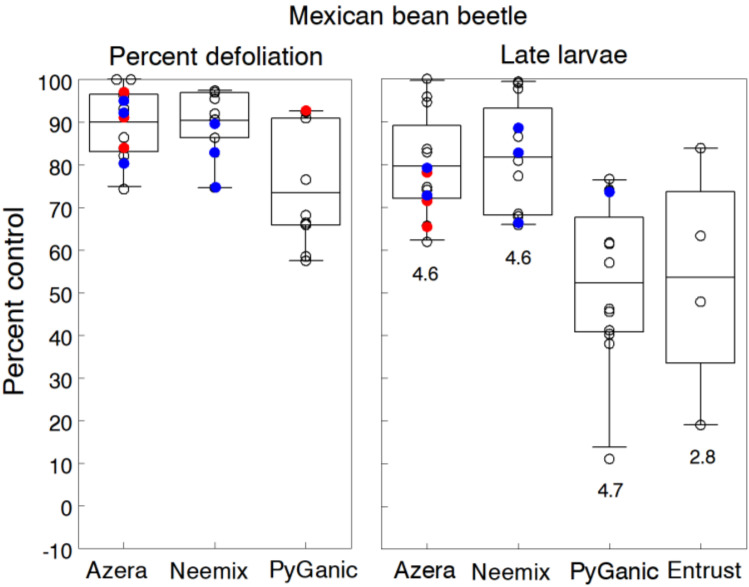
Percent control of Mexican bean beetles with weekly applications of Azera, Neemix, PyGanic and Entrust relative to the untreated control. Mean data are given for percent defoliation and late larvae (3rd and 4th instars) for nine trials on snap beans. Box-whisker plots show the 25% and 75% percentile range of percent control data and the horizontal line in each box is the overall mean control efficacy. Number below each plot indicates the average number of weekly applications. Application rates of Azera, Neemix, PyGanic and Entrust were 2.34 L/h, 1169 mL/h, 2.34 L/h and 71.2 g a.i./h respectively, except for individual trial data in blue and red indicating lower and higher rates, respectively. There were no significant differences in defoliation among treatments. Azera and Neemix provided significantly higher control of late larvae compared to the other insecticides.

**Figure 8 insects-11-00614-f008:**
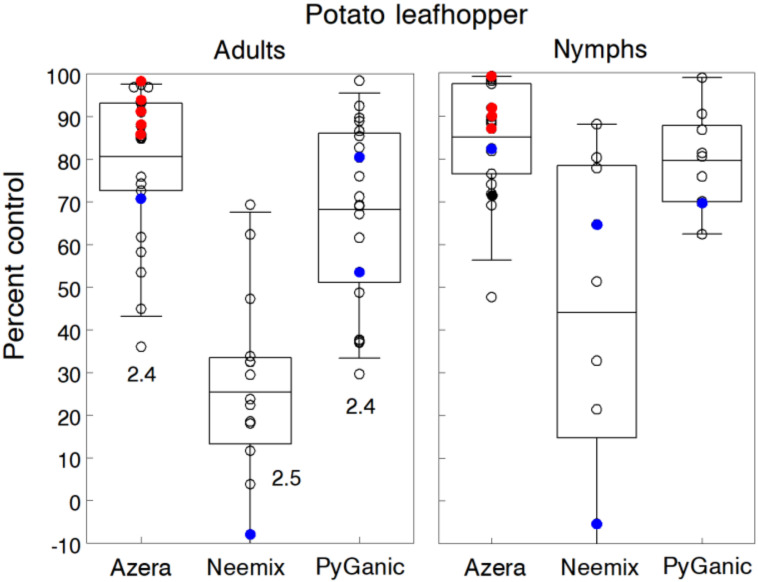
Percent control of potato leafhopper with weekly applications of Azera, Neemix and PyGanic relative to the untreated control. Mean data are given for adults and nymphs compiled from 4 trials on snap beans, 5 trials on alfalfa and 6 trials on potato. Box-whisker plots show the 25% and 75% percentile range of percent control data and the horizontal line in each box is the overall mean control efficacy. Number below each adult plot indicates the average number of weekly applications. Application rates of Azera, Neemix, PyGanic and Entrust were 2.34 L/h, 1169 mL/h, 2.34 L/h and 71.2 g a.i./h respectively, except for individual trial data in blue and red indicating lower and higher rates, respectively. Azera and PyGanic provided significantly higher control of adults compared to Neemix, but there were no significant differences in nymph control among treatments.

**Figure 9 insects-11-00614-f009:**
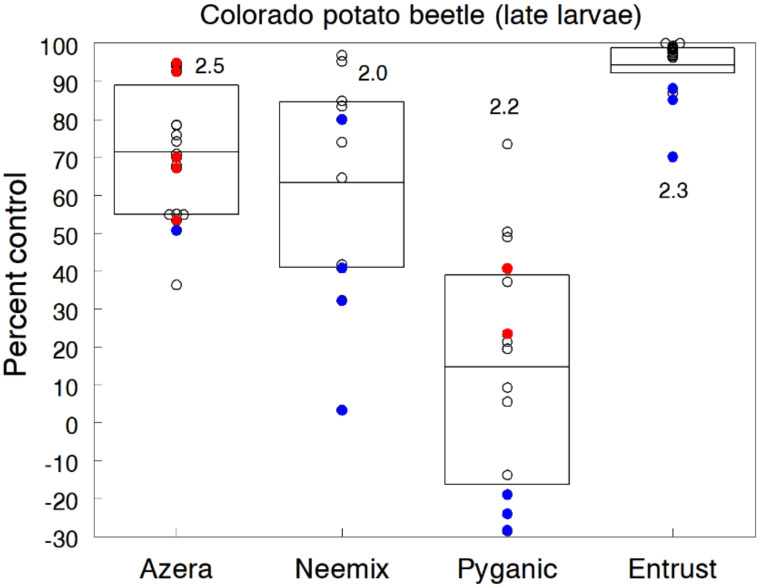
Percent control of Colorado potato beetle with weekly applications of Azera, Neemix, PyGanic and Entrust relative to the untreated control. Mean data are given for larvae (3rd and 4th instars) compiled from 14 trials on potato. Box-whisker plots show the 25% and 75% percentile range of percent control data and the horizontal line in each box is the overall mean control efficacy. Number above each plot indicates the average number of weekly applications. Application rates of Azera, Neemix, PyGanic and Entrust were 2.34 L/h, 1169 mL/h, 2.34 L/h and 71.2 g a.i./h respectively, except for individual trial data in blue and red indicating lower and higher rates, respectively. Azera, Neemix and Entrust provided significantly higher control of adults compared to PyGanic.

**Figure 10 insects-11-00614-f010:**
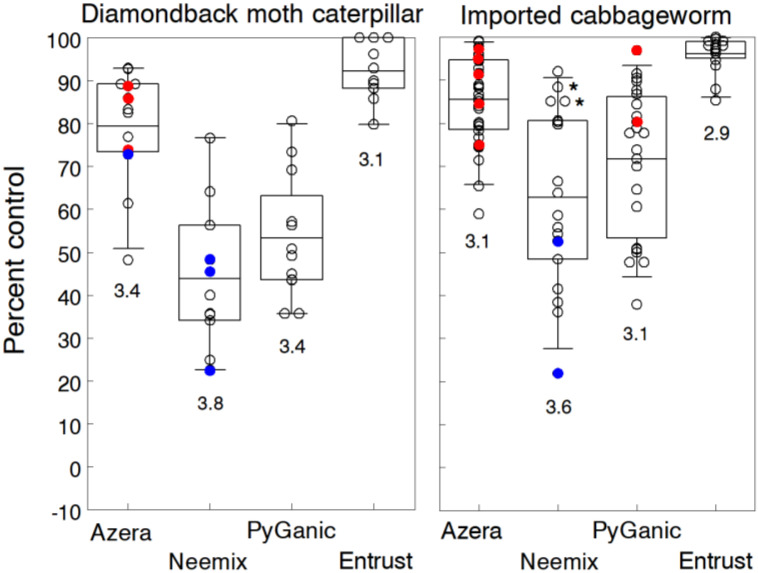
Percent control of cabbageworms with weekly applications of Azera, Neemix, PyGanic and Entrust relative to the untreated control. Mean data are given for diamondback moth caterpillar and imported cabbageworm compiled over 9 cabbage trials, 5 collard trials and 5 broccoli trials. Box-whisker plots show the 25% and 75% percentile range of percent control data and the horizontal line in each box is the overall mean control efficacy. Number below each plot indicates the average number of weekly applications. Application rates of Azera, Neemix, PyGanic and Entrust were 2.34 L/h, 1169 mL/h, 2.34 L/h and 71.2 g a.i./h respectively, except for individual trial data in blue and red indicating lower and higher rates, respectively. Azera and Entrust provided significantly higher control of both cabbageworms than Neemix and PyGanic.

**Figure 11 insects-11-00614-f011:**
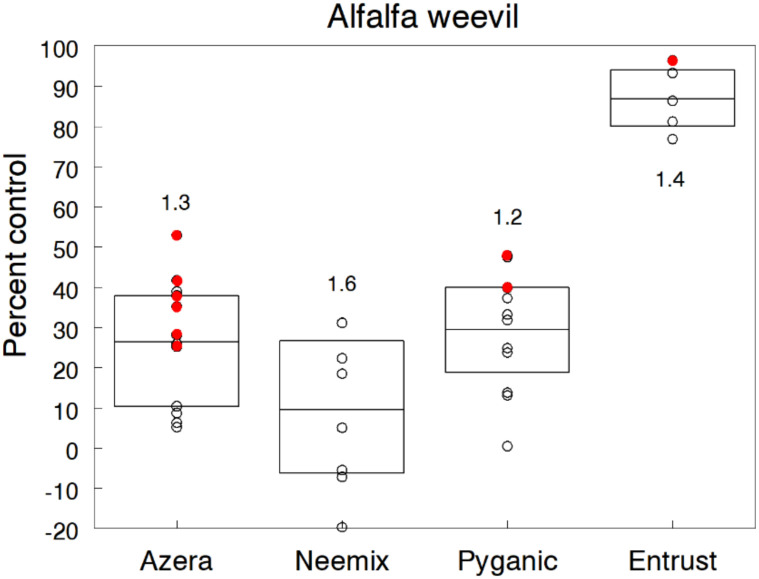
Percent control of alfalfa weevil with applications of Azera, Neemix, PyGanic and Entrust relative to the untreated control. Mean data are compiled from 10 trials on alfalfa. Box-whisker plots show the 25% and 75% percentile range of percent control data and the horizontal line in each box is the overall mean control efficacy. Number associated with each plot indicates the average number of weekly applications. Application rates of Azera, Neemix, PyGanic and Entrust were 2.34 L/h, 1169 mL/h, 2.34 L/h and 71.2 g a.i./h respectively, except for individual trial data in red indicating higher rates. Entrust provided significantly higher control compared to other insecticides.

**Figure 12 insects-11-00614-f012:**
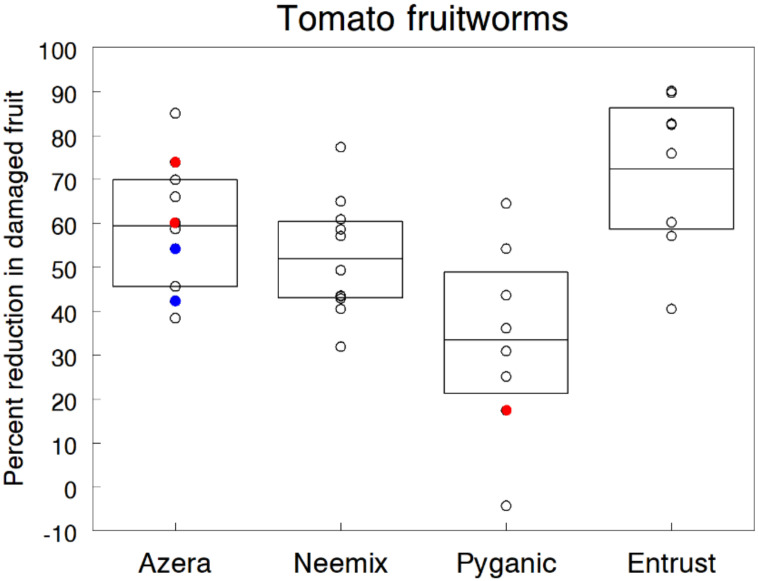
Percent control of tomato fruitworms with applications of Azera, Neemix, PyGanic and Entrust relative to the untreated control. Mean data are compiled from nine tomato trials, each receiving five weekly applications of each insecticide. Box-whisker plots show the 25% and 75% percentile range of percent control data and the horizontal line in each box is the overall mean control efficacy. Application rates of Azera, Neemix, PyGanic and Entrust were 2.34 L/h, 1169 mL/h, 2.34 L/h and 71.2 g a.i./h respectively, except for individual trial data in blue and red indicating lower and higher rates, respectively. There were no significant differences in control among treatments.

**Table 1 insects-11-00614-t001:** Insect pest/crop groups evaluated in the study.

Pest Group	Predominant Insect Species	Host Crops (*Cultivar*)	Number of Trials
Thrips (Thysanoptera: Thripidae)	Onion thrips, *Thrips tabaci*;	Onions (*Candy, Super Star*)	8
	Eastern flower thrips, *Frankliniella tritici*	Zinnia (*Benaryt’s Giant*)	10
Aphids (Hemiptera: Aphididae)	Green peach aphid, *Myzus persicae*	Collards (*Vates*), the Cabbage (var. *Blue Thunder*)	5
	Pea aphid, *Acyrthosiphon pisum*	Alfalfa (*Forage Queen*)	7
True bugs (Hemiptera: Pentatomidae, Coreidae)	Harlequin bug, *Murgantia histrionica*	Collards (*Vates*)	6
	Squash bug, *Anasa tristis*	Zucchini (*Payroll*)	4
	Brown marmorated stink bug, *Halyomorpha halys*; Brown stink bug, *Euschistus servus*	Tomato (*Mountain Plus*), Pepper (*Intruder*)	10
Flea beetles (Chrysomelidae: Coleoptera)	Eggplant flea beetle *Epitrix fuscula*; Tobacco flea beetle, *Epitrix hirtipennis*	Eggplant (*Dusky*)	11
Cucumber beetles (Chrysomelidae: Coleoptera)	Striped cucumber beetle, *Acalymma vittatum;* Spotted cucumber beetle, *Diabrotica undecimpunctata howardi*	Cucumber (*Dasher 1*) Zucchini (*Payroll*)	6
Japanese beetle	*Popillia japonica* (Coleoptera: Scarabaeidae)	Marigold (*Diamond Jubilee*) Basil (*Aroma 2*)	8
Mexican bean beetle	*Epilachna varivestis* (Coleoptera: Coccinellidae)	Snap beans (*Provider*)	9
Potato leafhoppers	*Empoasca fabae* Harris (Hemiptera: Cicadellidae)	Potato (*Kennebec*) Alfalfa (*Forage Queen*) Green beans (*Provider*)	15
Colorado potato beetle	*Leptinotarsa decemlineata* (Coleoptera: Chrysomelidae)	Potato (*Kennebec*)	14
Cabbage worms	Imported cabbageworm, *Pieris rapae* (Lepidoptera: Pieridae); Diamondback caterpillar, *Plutella xylostella* (Lepidoptera: Plutellidae)	Collards (*Vates*) Cabbage (*Blue Thunder*)	15
Alfalfa weevil	*Hypera postica* (Coleoptera: Curculionidae)	Alfalfa (*Repel ll*)	10
Fruitworms	Tomato fruitworm, *Helicoverpa zea*; yellow-striped armyworm, *Spodoptera ornithogalli*; variegated cutworm, *Peridroma saucia*	Tomato (*Mountain Plus*)	9
	Corn earworm, *Helicoverpa zea*	Sweet corn (*Prime Plus*)	6

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
