# Peer review of "Comparative Efficacy of Common Active Ingredients in Organic Insecticides Against Difficult to Control Insect Pests"

_insects, 2020, doi:10.3390/insects11090614_

Round 1

Reviewer 1 Report

Few suggestions to improve the text and few misprints to correct as listed below:

Line 54: change “spinose” with “spinosa”

Line 153-155: the statements should be supported by reference papers, if available, where the economic tresholds and the economic injury levels have been established for the different pest group investigated. These levels could be presented in tabular form.

Line 213: change “preformed” with “performed”

Line 615-621: change the format of this paragraph. It is not a figure caption!

Line 870: Finally, instead of “Fin him ally,”

Author Response

Line 54: changed spinose” to “spinosa”

Lines 153-154: reworded a sentence “high enough to cause economic damage.” We removed the word “level” we suggest that a threshold level is known for each pest group. However, there are very few established economic injury levels established for these pest groups in organic agriculture. The criteria for deciding when to applied the first application was basically based on the authors’ experience. We did however specifically describe a population or damage level that we considered enough to justify a treatment.

Line 213: changed “preformed” to “performed”

Lines 615-621: changed the format of this section to a normal paragraph of the text.

Line 870: changed “Fin him ally,” to “Finally,”

Reviewer 2 Report

The manuscript by Dively et al. summarizes the results of 153 (!) field efficacy trials conducted over 14 years to investigate the efficacy of four insecticides approved for use in organic production: Entrust, Azera, PyGanic and Neemix.  This is a unique dataset that will be a valuable resource for scientists and pest managers interested in organic production.  Despite its length, the manuscript was easy to review because the sections were formatted in a similar manner and the manuscript was well-written.  I have only a few minor comments.

Given that over 150 trials were conducted, the materials and methods are naturally open to some criticism because of differences among trials in rates, formulations, application equipment (nozzles, etc.), production practices, etc.  Nonetheless, the trials appeared to be conducted carefully and the methods seemed relatively consistent among trials.  The section on “Insect sampling” was not as well-referenced as it should have been.  In the “Data analysis”, it is not clear whether sampling date was used as both a fixed effect and a repeated measure.  The sentences describing the calculation of “percent control” were a little ambiguous – perhaps a formula would help, because I am not sure what is meant by “change in insect density”.  Also, throughout the manuscript, the phrase “weekly applications” is used.  This phrase was usually used to indicate the number of weeks that an application was made at weekly intervals, but there seemed to be a couple of cases in which it referred to the number of applications made in a week (e.g., line 383).  Or am I mistaken?

Were conventional insecticides ever included in the trials for the sake of comparison?

Line 339, replace “tolerate” with “tolerant”

Figure 6, legend, delete the fourth sentence

Figure 9, problem with formatting above the Figure

Line 748, replace “Most predominate”  with “The predominant species”

Author Response

The section on “Insect sampling” was not as well-referenced as it should have been. All the sampling methods described for each pest group are commonly used in entomological field studies. None were modified from the standard sampling approaches. Thus we don’t see the need to reference standard methods.

In the “Data analysis”, it is not clear whether sampling date was used as both a fixed effect and a repeated measure. The model included insecticide treatment, sampling date, and the interaction as fixed factors, replicate block as a random effect, and sampling date as a repeated measure. This sentence in section 2.5 clearly states that sampling date was used both as a fixed factor and a repeated measure.

The sentences describing the calculation of “percent control” were a little ambiguous. We reworded the following to clearly explain how percent control was calculated. “In both cases, we averaged the post-treatment data on insect density over all sampling dates of each trial and calculated the mean percent control for each insecticide, using Abbott’s formula [(control density - treatment density)/control density*100].”

Comment about the phrase “weekly applications”  We stated in the methods section that applications in all trials were made on a weekly basis and most figures indicate the average number of applications averaged over all trials for each pest group. There were a few trials were applications for made at shorter intervals but the text explains these special cases in detail. However, we did reword line 384 as follows: “applied either 2 or 3 times on a weekly basis.”

Yes, a few trials included standard conventional insecticides but we did not think it was relevant to compare their efficacy with the organic products.

Line 339: replaced “tolerate” with “tolerant”

Figure 6, legend, deleted the fourth sentence. DONE

Lines 615-621: changed the format of this section above figure 9 to a normal paragraph of the text.

Line 748: Replaced “Most predominate”  with “The predominant species”